# The Evaluation Model on an Application of SROI for Sustainable Social Enterprises

**Dong-Joo Kim [1] and Yong-Seung Ji [2,*]**

[1] Department Rehabilitation Studies, Woosuk University, Wanju-gun 55338; Korea; ju7055@naver.com
[2] Department Divisions of Liberal Arts, Woosuk University, Wanju-gun 55338, Korea
* Correspondence: enerji1008@gmail.com

**Abstract:** The evaluation of social enterprises has been criticized for not being able to reflect the positive social aspects of the company through the financial evaluation. SROI (Social Return on Investment) is a social concept applied to the measurement of economic return on investment that aims to measure the social added value of social enterprises and reflect them in their performance. It is necessary to research for the provision of support and management for sustainable social enterprises, and to prepare a method for evaluating social enterprise by applying SROI. The Delphi 1st and 2nd surveys for the development of evaluation model for social enterprises using SROI were conducted by 50 social enterprise CEOs and experts. To produce the results of this study, the SPSS 20.0, AMOS 24 and Expert Choice 11 programs were used and the pairwise comparison analysis method was performed to analyze importance and priority. The results of the Delphi and AHP (Analytic Hierarchy Process) analysis showed that employment was the most important factor in social enterprises with the highest share of newly hired personnel. Employment-type social enterprises have the highest priority in terms of employment, income (income increase for vulnerable workers), and community contribution (affordable of social services), while the social service type was in order of employment, community contribution, and income (income increase for the vulnerable workers). On the other hand, the mixed type was employment (newly hired personnel), income (income increase of vulnerable workers), employment (social work participants' switch to similar work after contract expiration), and community contribution (affordable of social services). This study makes efforts to form social capital by raising the public's awareness of social value with efficient management through various evaluations of social enterprises and the emergence of various social enterprises. This study also emphasizes the need to better understand social enterprises as a multi-scholar and multi-dimensional organization that includes a multi-faced mechanism of social, economic, and environmental community development, away from understanding social enterprises as a specific business model.

**Keywords:** SROI; sustainable social enterprises; delphi; AHP

## 1. Introduction

As interest in social enterprises has been rising, the issue of sustainable social enterprise has been strongly raised as the government has a strong will to foster social enterprises. As the number of social enterprises increases, there is a need for management and follow-up measures for social enterprises, and there is a constant question as to whether sustainable social organizations in a broad sense will have a future [1–3]. As a measure to secure the sustainability of social enterprises, the government intends to strengthen its monitoring and evaluate the social enterprises through government-led measures. However, the evaluation of social enterprises has been criticized because it does not reflect the positive aspect of social enterprise through financial evaluation [4]. SROI is a social concept applied

to the measurement of economic return on investment, which is to measure the social added value of social enterprises and reflect them in their performance. There are no examples of local governments applying SROI to social enterprises. Even in the country, only research on the direction of the SROI is being conducted, and there is no discussion on how to apply it.

The ROI (Return on Investment), which is currently being measured in the evaluation system for social enterprises, measures only the economic value added to the economic benefits minus the economic costs. It can be understood as the recovery rate of investment. Future cash flows are expressed as a percentage of investment. This is expressed as the ratio of the present value of the number of returns to the present value of the investment in terms of the enterprise, which is not suitable as an evaluation tool of social enterprise. While the realization of social objectives is emphasized as an important role of social enterprises, the lack of an accurate evaluation of these will not be accurately conveyed. This can be seen as a compelling social enterprise to achieve both social goals and realization of benefits.

In the case of social enterprises in Europe, the study of social enterprises was mainly influenced by the economic ripple effects of social enterprises and their impact on regional development [5,6]. On the basis of this, research on the social value of social enterprises has begun on the fact that other evaluation tools must be prepared for accurate evaluation of social enterprises. The effort to measure the social contribution of social enterprises as a measure of their return on investment has begun. Various evaluation methods such as BSC (Balanced Scorecard) and DEA (Data Envelopment Analysis) have been studied as tools for performance evaluation. Recently, SROI centered on REDF (Robert Enterprise Development Fund). The value of the enterprise has begun to become more active [7–10].

The most important goal of social economic organizations is to increase sustainability because of the nature of social enterprises included in the social economy area [11–13]. The social enterprise sustainability criteria are summarized in terms of process and structure, human resources, finance, governance, performance measurement, and market recognition [14–16]. In particular, SROI, as a measure of performance measurement, is a social concept applied to the measurement of economic return on investment and is intended to measure the social added value of social enterprise and to reflect it on its performance.

The introduction of the SROI concept is unfamiliar and has not yet been applied at the national level. The evaluation of the social enterprise up to now is carried out by the Ministry of Labor in Korea and the evaluation of the concept of ROI excluding the social concept. In the early stage of social enterprise development and lack of understanding about social enterprises, evaluation of ROI concept can be a factor to devalue the value of social enterprise. Therefore, it is necessary to provide sustainability of social enterprise through accurate assessment of social enterprise and to provide opportunities for social enterprise and community to assimilate through publicity. Even the government is only conducting research on the direction of SROI, and discussion on how to apply it is not proceeding. This study is needed to prepare a plan for the support and management of social enterprises from the viewpoint of the government. The purpose of this study is to establish a method to evaluate Korean social enterprises by applying SROI. Therefore, in this study, as a cornerstone for developing a standardized approach for measuring social values that can be applied simultaneously to not only social enterprises but also general enterprises, we assessed approaches to measuring social value centered on SROI and explored development directions for their standardization.

## 2. Literature Review

### 2.1. Application of SROI

Social enterprises create economic and social values, which are outcomes in the market and society, and economic values can be measured objectively by financial statements. However, social values are difficult to measure, and subjective judgment can be involved even if they can be measured. In an attempt to solve these difficulties, there have been many studies for measuring social value

of social enterprises. Representative indicators are proposed by REDF, recently internationally standardized indicators by The SROI Network through NEF (The New Economics Foundation) is a SROI [17]. SROI was developed in 2000 by the US private foundations REDF and Jed Emerson and is a measurement tool developed to measure the socio-economic value of investments or contributions to social enterprises [18]. SROI is the ratio between the value created by the operation of a social enterprise and the investment required to achieve its effects. SROI is a framework for observing social influence, and the work of converting it to monetary value plays an important role, but this is not all [19]. SROI is not about money but is about value. Money is simply a unit, and a useful and widely accepted means of exchanging value. In the context of a business plan containing a lot of information beyond financial design, SROI is more than just numbers [20]. Ultimately, SROI is a tool to convert into data that can be understood by a wide range of stakeholders, including those who want to influence the social value created, and those who provide the support necessary for success. In other words, SROI is the ratio of the NPV (net present value) of the result to the NPV of the investment and measures the value of the result generated against the cost of achieving that result [19].

### 2.2. SROI and Measurement of Social Value

Looking closely at the concept of SROI, it applied the concept of ROI which is an indicator of general corporate accounting and is generated from investments such as capital investments by investors outside the company, facility investments inside the company, and R&D investments. It is a representation of future cash flows as a ratio and is an indicator for evaluating the feasibility and final investment performance of an alternative investment. In general, ROI deals only with economically generated returns, so it is effectively an EROI (Economic Return on Investment). SROI can be seen as a ratio of the future social (monetary) value resulting from investments by applying the concept of EROI. Since the investment amounts are the same in the EROI and SROI, the denominators are the same, so they can be added together. The sum of EROI and SROI is called BROI (Blended ROI, socio-economic return on investment, SEROI), and BROI is commonly referred to as SROI [18].

Since ROI analyzes the future cash flows of a company for 5 to 10 years, SROI is also evaluated through the forecast results of cash flows and social performance on investments of social enterprises for 5 to 10 years. When evaluating past performance through investments, the performance from the time of investment to the present is calculated and collected. Since it is difficult to predict the period when investment effects will occur, SROI including both past performance and future projections can be calculated. However, there is a problem in evaluating short-term performance on a yearly basis, and SROI reflects the position of social venture capital investing capital mainly in social enterprises, and the key components of SROI can be composed of investment, investment return, economic return, social return, social benefit, social cost, and so on.

Economic profit is net income plus interest expense, which can be easily calculated on the financial statements. Social profit refers to the profits of stakeholders, and by applying the concept of profit, social profit can be defined as 'the extent to which social assets have grown due to the investment' [9].

As such, the socio-economic value that arises from real economic activity but cannot be measured by economic accounting is called the external effect and the positive externalities that lead to the increase of social assets means the social benefits, on the other hand, the negative externalities that lead to a reduction in social assets means social costs. Since social profits are social benefits minus social costs, social profits can be seen as the sum of all external effects. Social benefits refer to stakeholders' benefits, that is, the increase in assets held by stakeholders and the increase in opportunity benefits.

### 2.3. Application of SROI and Sustainability

SROI describes the changes made by measuring social, environmental and economic outcomes, and expresses and assigns monetary values to symbolize them. This can be estimated by cost-benefit analysis. SROI measurement process should focus on the perspective of changes that occur or are predicted by various stakeholders as a result of the activity and can be divided into six stages as

follows. First, it is important to establish a clear definition of the scope of SROI analysis and the scope and method of stakeholders' participation in the analysis process. Second, in the mapping phase, impact map or theory of change can be developed to confirm the relationship among inputs, outputs, and outcomes through stakeholders' engagement. Third, as the process of identifying results and giving, it is the process of confirming the evidence that the result actually happened and of giving social value to the evidence. Fourth, confirmation of impact is the process of determining whether such a change has occurred with the confirmed evidence and the monetized value of the result or recognizing it as a result of other factors and excluding it. Fifth, SROI calculation step combines all the identified social benefits, deducts negatives, and compares the results of investments. It is also possible to conduct a review through sensitivity analysis on the calculated results. Finally, it is an easy to overlook, but very important, "reporting, application, and internalization" process that shares the results of SROI measurements with stakeholders, responds to their questions, and internalizes them through exemplary process and verification of reports [9]. SROI measurement is available in the REDF method of US and the NEF method of UK. While the former emphasizes value calculation, the latter specifies stakeholders' prior identification and sensitivity analysis, but both follow substantially the same steps [21].

The results of SROI performance measurement on the social value of social enterprises can be used as useful indicators in various areas related to social enterprise management. The application of SROI for sustainability of social enterprises is as follows. First, it can be used as a supplementary data for unstable management performance in evaluating management performance of social entrepreneurs, as well as it can be an advantage to provide an opportunity to interfere with management performances of various stakeholders on social entrepreneurs of social enterprises with low SROI measurement results. Second, social enterprises operate by receiving subsidies from the government through various channels. SROI can be a valuable data that can be used as a performance evaluation indicator for government support, and can be used to establish differentiated support policies, such as providing incentives for social enterprises with high SROI performance. Third, with the growth of the social enterprise sector, interest in social enterprises has increased in various social investors. SROI measurement results can provide useful information when making investment decisions in social enterprises. Finally, for the sustainability of social enterprises, public policy development agencies can provide useful information for SROI in recognizing the social value of policy alternatives. SROI can provide useful information in recognizing the social value of policy alternatives in public policy development agencies [9].

## 3. Materials and Methods

According to the type of social enterprise, it is classified into employment type, social service type, and mixed type, and in each case, the evaluation indicators are examined and their applicability is determined. This study secures possible indicators for the evaluation indicators presented through case studies at home and abroad. Due to the various characteristics of social enterprises, various indicators can be produced and analyzed for their applicability.

### 3.1. Research Target and Collecting Survey

In this study, the Delphi 1st and 2nd surveys for the development of evaluation model for social enterprises in Korea using SROI were conducted from Sep. 19 to 27, 2017 by 50 social enterprise CEOs and experts. A total of 29 respondents from the second survey were interviewed by 11 CEOs of social enterprises, 1 public employee related to social enterprises, 12 college professors, and 5 social enterprise experts. They were interviewed through structured questionnaires and emailed.

### 3.2. Research Methods

In order to carry out the research, we first reviewed the literature on SROI and AHP (Analytic Hierarchy Process). Through literature review, we understand the contents of existing research and indirectly raise the possibility of applying it to social enterprise. Next, we review the application process of SROI and analyze exiting SROI cases. Through case analysis, evaluation indicators

are secured by division and the evaluation indicators are sorted by category. The sectoral pool of evaluation index on social enterprise is secured through literature review of the existing SROI. Delphi survey is conducted on the pools for each sector, and key indicators are selected through additional evaluation indicators and additional processes. Finally, AHP is used to calculate inter-sector weights. In addition, through interviews with the CEOs of social enterprises, the possibility of application of SROI is increased by identifying problems and solution for SROI application.

### 3.3. Data Analysis

In this study, the lowest level was constructed around the indicator items derived through Delphi, and the homogeneous element was arranged around the lower level. First, in the 1st Delphi survey, items were selected based on the responsiveness average of respondents to select areas and items suitable for evaluating the economic value of social enterprises. Second, the 2nd Delphi survey reevaluates the appropriateness to each social value area and sub-items in order to verify social values and indicators modified by the 1st Delphi survey. At this time, the panel was asked to refer to the results of the 1st survey. In addition to evaluating role areas and subcategories, comments were added for each area and item. Third, social value areas and indicators of social enterprises modified through this process were finally identified. Finally, the 1st and 2nd Delphi surveys were used to calculate the indicators and weights were selected for them. Weights were calculated using AHP to assess the most important factors and weights to evaluate specific gravity in social enterprises. To produce the results of this study, the SPSS 20.0, AMOS 24 and Expert Choice 11 programs were used and the pairwise comparison analysis method was performed to analyze importance and priority. The level $\alpha$ to which the Type I error will be made is set at 0.5.

## 4. Results

The areas of social valuation and indicators were derived through previous studies on social valuation, literature reviews, and researchers. Statistical analysis is required to verify the validity of the derived items. For this study, Delphi survey was conducted for the professors, social enterprise officials, and social enterprise workers. Based on the statistical analysis results, after removing the less significant items, the social value areas and items for evaluating social value were finalized. The result is as follows.

### 4.1. Characteristics of the Subjects

According to the characteristics of respondents based on the results of the final survey, 82.8% were male (24 persons) and 17.2% were female (5 persons). In the age group, 17.2% were 30–39 years old (5 persons), 55.2% were 40–49 years old (16 persons), and 27.6% were over 50 years old (8 persons). Lastly, in the work group, college professors were 41.4% (12 persons), social enterprise workers were 37.9% (11 persons), public employee related to social enterprise was 3.4% (1 person), and other social enterprise experts were 17.2% (5 persons). In general, they are operating social enterprises or experts are in their forties. In terms of final education, the percentage of graduate school was the highest, and college graduates were next, with more opportunities for start-ups toward higher education. In terms of occupations, the number of workers in social enterprises was the highest, followed by other social enterprise experts, university professors, and public employee (Table 1).

### 4.2. Characteristics of Correlation among Major Variables

In order to verify the revised social value domains and indicators through the 1st Delphi survey, the 2nd Delphi survey was to reevaluate the suitability of each social value domain and items. The panel added comments on each area and items, as well as evaluating social value domain and items. Finally, the removed or corrected items were identified for the revised social value domain and indicators of social enterprise. The results of analyzing the validity and reliability of the panel's evaluation items on the 2nd Delphi survey are as follows.

**Table 1.** Characteristics of the Subjects.

| Variables | Category | 1st Survey | | 2nd Survey | |
|---|---|---|---|---|---|
| | | Frequency | Percentage | Frequency | Percentage |
| Gender | Male | 28 | 82.4 | 24 | 82.8 |
| | Female | 6 | 17.6 | 5 | 17.2 |
| Age | 30–39 years old | 5 | 14.7 | 5 | 17.2 |
| | 40–49 years old | 19 | 55.8 | 16 | 55.2 |
| | Over 50 years old | 10 | 29.4 | 8 | 27.6 |
| Academic Status | Under college | 3 | 8.8 | 1 | 3.4 |
| | University | 9 | 26.5 | 8 | 27.6 |
| | Graduate school (attending) | 22 | 64.7 | 20 | 69.0 |
| Jobs | College professors | 13 | 38.2 | 12 | 41.4 |
| | Social enterprise workers | 14 | 41.2 | 11 | 37.9 |
| | Public employee related to social Enterprise | 1 | 2.9 | 1 | 3.4 |
| | Other experts related social enterprise | 6 | 17.6 | 5 | 17.2 |
| | **Total** | 34 | 100.0 | 29 | 100 |

### 4.2.1. Social Value Domains

In this study, to verify the validity of the evaluation items for the 2nd survey, an estimation was made by the correlation between individual items and the total score, and the reliability of the items was calculated by Cronbach's $\alpha$ coefficient to estimate the degree of agreement between the items. The results are shown in Table 2. The correlation between the items and the total scores in the social value domain was 0.426–0.711. According to the correlation coefficient between the total score and each item, the 7th item of 'self-esteem recovery and increase' did not show a statistically significant correlation with the total score. The 1st item of 'job creation', the 9th item of 'protection for safety', and the 10th item of 'the government's budget reduction' were significant at the significance level of 0.05. The 10 items of the savings were significant at the significance level of 0.05, the 2nd item of 'increase of worker's income', the 3rd item of 'increase of income of workers'/service users' family', the 4th item of 'transitional job', the 5th item of 'enhancing worker's job ability', and the 6th item of 'strengthening social network and participants' self-development', the 8th item of 'community contribution', and the 11th item of 'improvement of health level' were significant at the level of 0.01. Also, the Cronbach's $\alpha$, which is a reliability coefficient indicating the agreement between the evaluation items, was 0.728, showing high reliability. In addition, since there were no sub-items related to 'strengthening social networks' in the areas of 'strengthening social networks and participant self-development', we also suggested opinions on index correction by 'participant self-development'.

**Table 2.** Validity and Reliability of Social Value Domains in Social Enterprises.

| Item Number | Item Contents | 2nd Survey | | | | Remarks |
|---|---|---|---|---|---|---|
| | | M | SD | Corr. | Cronbach's | |
| 1 | Job Creation | 4.61 | 0.497 | 0.426 * | 0.718 | |
| 2 | Workers' Income Increase | 4.18 | 0.612 | 0.711 ** | 0.675 | |
| 3 | Income Increase of Worker/Service Users' Family | 4.04 | 0.637 | 0.706 ** | 0.675 | |
| 4 | Transitional Job | 3.89 | 0.629 | 0.489 ** | 0.714 | |
| 5 | Enhancing Workers' Job Ability | 3.89 | 0.629 | 0.622 ** | 0.691 | |
| 6 | Strengthening Social Network & Participant Self-Development | 3.82 | 0.548 | 0.584 ** | 0.697 | |
| 7 | Self-esteem Recovery & Increase | 3.72 | 0.744 | 0.280 | 0.759 | deleted |
| 8 | Community Contribution | 4.61 | 0.567 | 0.521 ** | 0.707 | |
| 9 | Protection for Safety | 3.39 | 0.567 | 0.441 * | 0.719 | |
| 10 | Government's Budget Reduction | 3.82 | 0.612 | 0.318 * | 0.726 | |
| 11 | Improvement of Health Level | 3.82 | 0.723 | 0.646 ** | 0.688 | |
| | | | | | **Total Cronbach's =0.728** | |

Confidence Level: * $p < 0.05$, ** $p < 0.01$.

### 4.2.2. Items of Social Enterprises' Measurement

The correlation between the items and the total score of the social value index was 0.256–0.812. As a result of the correlation coefficient between total score and each item, the 10th item of 'increasing self-esteem through vocational activities' did not show a statistically significant correlation with total score. The 1st item of 'newly hired personnel', the 2nd item of 'income increase for vulnerable workers', the 3rd item of 'income increase of workers/service users' family through economic activities', the 6th item of 'degree of technical competence through vocational activities', the 9th item of 'providing social training for workers', the 14th item of 'budget reduction through consignment management of social welfare', and the 17th item of 'reduction of family's care-cost' were significant at the significance level of 0.05. The 4th item of 'social work participants' switch to similar work after contract expiration', the 5th item of 'certification through vocational activities', the 8th item of 'providing cultural programs for workers', the 11th item of 'affordable social services', the 12th item of 'free provision of the social services', the 13th item of 'reduction of safety accidents in social enterprises', the 14th item of 'budget reduction through consignment management of social welfare services', and the 15th item of 'reduction of use of tertiary care institutions' were significantly higher at the level of 0.01. Also, the value of Cronbach's α, which is the coefficient of confidence indicating the degree of agreement among the evaluation items, was 0.863, showing high reliability. However, the Cronbach's α value was 0.866, which is higher when removing the 10th item of 'increasing self-esteem through vocational activities'. Therefore, it is considered that it is appropriate to exclude this item (Table 3).

**Table 3.** Validity and Reliability of Social Value Items in Social Enterprises.

| Item Number | Item Contents | 2nd Survey | | | | Remarks |
|---|---|---|---|---|---|---|
| | | **M** | **SD** | **Corr.** | **Cronbach's** | |
| 1 | Newly Hired Personnel | 4.44 | 0.641 | 0.408 * | 0.861 | |
| 2 | Income Increase for Vulnerable Workers | 4.33 | 0.555 | 0.256 * | 0.858 | |
| 3 | Income Increase of Workers'/Service Users' Family Through Economic Activities | 4.15 | 0.602 | 0.484 * | 0.854 | |
| 4 | Social Work Participants' Switch to Similar Work After Contract Expiration | 4.04 | 0.898 | 0.693 ** | 0.853 | |
| 5 | Certification Through Vocational Activities | 4.07 | 0.730 | 0.578 ** | 0.857 | |
| 6 | Degree of Technical Competence Through Vocational Activities | 4.26 | 0.526 | 0.473 * | 0.860 | |
| 7 | Family Counseling & Free Education | 3.93 | 0.874 | 0.672 ** | 0.852 | |
| 8 | Providing Cultural Programs for Workers | 4.22 | 0.801 | 0.808 ** | 0.840 | |
| 9 | Providing Social Training for Workers | 3.85 | 0.718 | 0.439 * | 0.857 | |
| 10 | Increasing Self-esteem Through Vocational Activities | 4.07 | 0.651 | 0.088 | 0.866 | Deleted |
| 11 | Affordable Social Services | 4.41 | 0.572 | 0.555 ** | 0.855 | |
| 12 | Free Provision of Social Services | 4.04 | 0.706 | 0.422 ** | 0.857 | |
| 13 | Reduction of Safety Accidents in Social Enterprises | 3.59 | 0.797 | 0.633 ** | 0.854 | |
| 14 | Budget Reduction Through Consignment Management of Social Welfare Services | 3.89 | 0.698 | 0.397 * | 0.860 | |
| 15 | Reduction of Use of Tertiary Care Institutions | 3.96 | 0.898 | 0.812 ** | 0.837 | |
| 16 | Reduction of Hospitalization Days | 3.37 | 0.742 | 0.464 * | 0.860 | |
| 17 | Reduction of Family's Care-cost | 3.37 | 0.742 | 0.424 * | 0.859 | |
| | | | | | **Total Cronbach's = 0.863** | |

Confidence Level: * $p < 0.05$, ** $p < 0.01$.

### 4.3. AHP System Configuration

In this study, the lowest level was constructed around the indicator items derived through Delphi, and the homogeneous elements were arranged around the constructed level. The next level was arranged based on the relationship between the area of social value evaluation derived from Delphi and the lower level. At the top level, the elements are clustered and arranged around the elements, which arranged at the middle level. The first-level hierarchical structure was based on the relationship between levels [22] and was centered on clusters of similar elements. According to Saaty, the only limitation in hierarchical structure is that all elements at one level should relate to some at adjacent

levels [22]. The AHP system configuration were organized through expert advices, such as researchers, AHP experts, and academia, based on the hierarchies as shown in Figure 1.

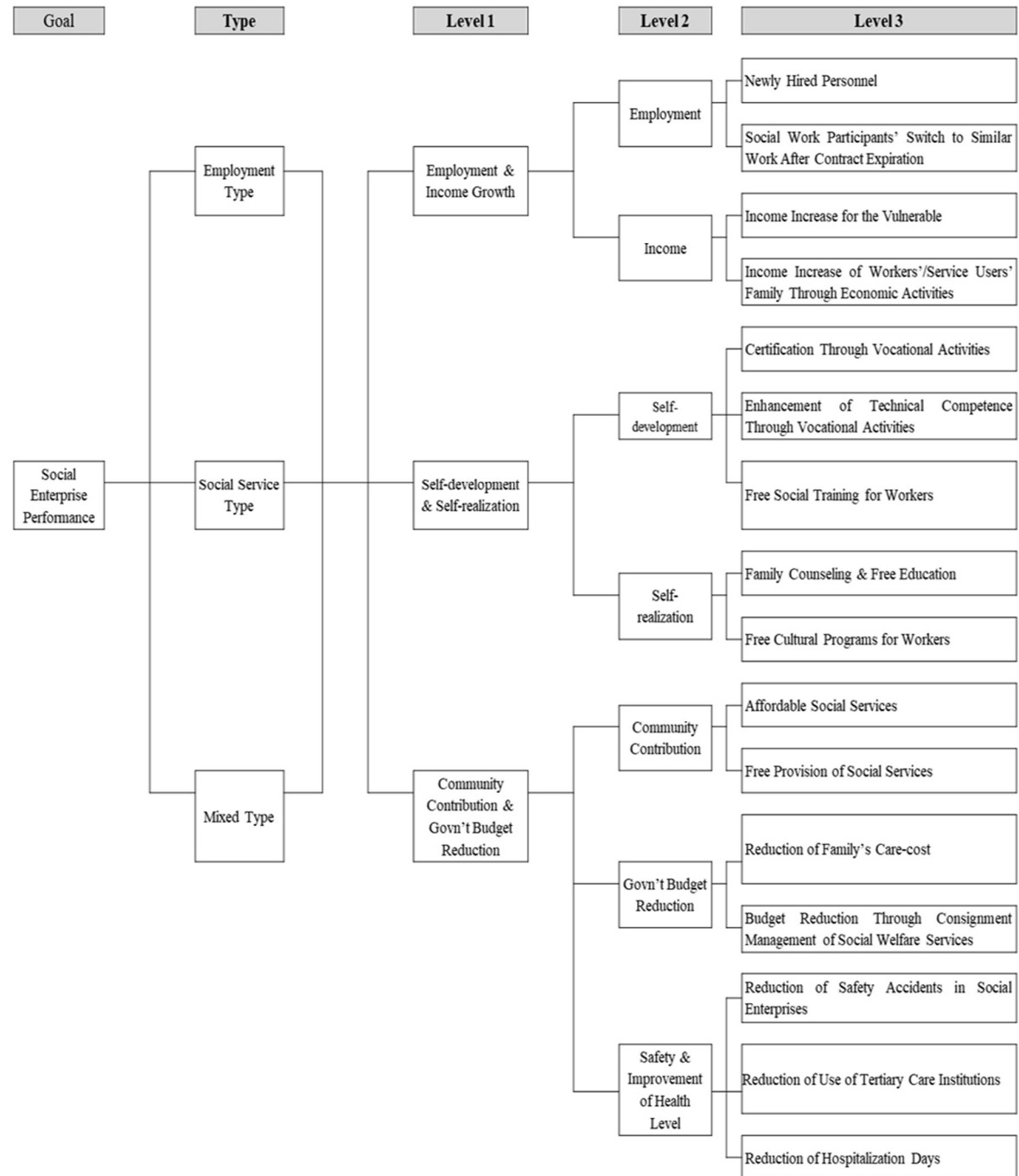

**Figure 1.** AHP System.

### 4.4. Estimation of Weights by Sector through AHP Analysis

In the AHP importance analysis process used in this analysis, only the response data that was identified as less than 0.2 through the consistency ratio among the responses of the AHP survey subjects were validated and used for AHP data analysis. The consistency is a criterion for verifying the consistency of responses in the AHP technique analyzed through pairwise comparisons, and the CI (consistency index), and the CR (consistency ratio) can be obtained by the following equation. According Saaty, AHP analysis calculates the CI and then uses the RI (random index) to calculate

the CR. It is a concept to measure the difference between the extracted weight and the response. To perform consistency analysis, first, we find the principal Eigenvalue ($\lambda$ *max*), second, the CI should be obtained using $\lambda$ *max*. Third, we obtain the CR with CI. With this CR, it is determined whether or not the coincidence [22]. Saaty showed that for consistent reciprocal matrix, the largest Eigen value is equal to the size of comparison matrix, or $\lambda$ *max* = *n*. Then he proved a measure of consistency, called Consistency Index as deviation or degree of consistency using the following Formula (1).

$$\text{Consistency Index(CI)} = \frac{\lambda\ max - n}{n - 1}, \lambda\ max\ \geq n \tag{1}$$

Saaty proposed that we use this index by comparing it with the appropriate one [22]. The appropriate Consistency index is called Random Consistency Index (*RI*). Then, he proposed what is called Consistency Ratio, which is a comparison between Consistency Index and Random Consistency Index, or in Formula (2).

$$\text{Consistency Ratio(CR)} = \left(\frac{CI}{Random\ Consistnecy\ Index}\right) \times 100\% \tag{2}$$

If the value of Consistency Ratio is smaller or equal to 10%, the inconsistency is acceptable. If the Consistency Ration is greater than 10%, we need to revise the subjective judgment. This equation was used to determine the consistency of respondents for Level I and Level II. In AHP, there are two main ways to combine group estimates. In other words, it can be divided into 'group evaluation method' and 'numeric integration method' [22]. In this study, we used the commonly used numerical integration method and integrated weight estimation method using arithmetic mean among its methods.

As a result of the AHP evaluation, the most important factor in social enterprises was employment, with the highest proportion of newly hired personnel. In employment type, the proportion was 55.9%, social service type was 23.4%, and mixed type 34.5%. Employment-type social enterprises are in the order of employment, income (income increase for vulnerable workers), and community contribution (affordable social services), while social service-type social enterprises have employment, community contribution, income (income increase for vulnerable workers). On the other hand, mixed (employment- and social service-type) social enterprises show that employment (newly hired personnel), income (increased income of vulnerable workers), employment (social work participants' switch to similar work after contract expiration), and community contribution (affordable social services) are appeared in the Table 4. Overall, employment, community contribution, and income were considered as the most important factors. Therefore, items such as employment, income, and community contribution could be used to assess social enterprises.



**Table 4.** Items and Weights by Type of Social Enterprises.

| Social Value | | | Employment-Type | | Social Service-Type | | Mixed-Type | |
|---|---|---|---|---|---|---|---|---|
| Level 1 | Level 2 | Level 3 | Weights | Rank | Weights | Rank | Weights | Rank |
| Employment & Income Increase | Employment | Newly Hired Personnel | 0.559 | 1 | 0.234 | 1 | 0.345 | 1 |
| | | Social Work Participants' Switch to Similar Work After Contract Expiration | 0.109 | 2 | 0.125 | 2 | 0.093 | 3 |
| | Income | Income Increase for Vulnerable Workers | 0.088 | 3 | 0.076 | 5 | 0.132 | 2 |
| | | Income Increase of Workers'/Service Users' Family Through Economic Activities | 0.019 | 8 | 0.043 | 10 | 0.058 | 5 |
| Self-development & Self-esteem | Self-development | Certification Through Vocational Activities | 0.030 | 6 | 0.045 | 9 | 0.058 | 6 |
| | | Technical Competence Through Vocational Activities | 0.035 | 5 | 0.066 | 6 | 0.055 | 7 |
| | | Providing Social Training for Workers | 0.016 | 10 | 0.042 | 11 | 0.027 | 11 |
| | Self-esteem | Family Counseling & Free Education | 0.012 | 12 | 0.049 | 7 | 0.038 | 9 |
| | | Providing Cultural Programs for Workers | 0.016 | 9 | 0.047 | 8 | 0.032 | 10 |
| Community Contribution & Govn't Budget Reduction | Community Contribution | Affordable Social Services | 0.049 | 4 | 0.082 | 4 | 0.069 | 4 |
| | | Free Provision of Social Services | 0.025 | 7 | 0.087 | 3 | 0.040 | 8 |
| | Govn't Budget Reduction | Reduction of Family's Care-cost | 0.014 | 11 | 0.034 | 12 | 0.015 | 13 |
| | | Budget Reduction Through Consignment Management of Social Welfare Services | 0.011 | 13 | 0.030 | 13 | 0.018 | 12 |
| | Safety & Increase Health Level | Reduction of Safety Accidents in Social Enterprises | 0.008 | 14 | 0.016 | 14 | 0.011 | 14 |
| | | Reduction of Use of Tertiary Care Institutions | 0.005 | 15 | 0.013 | 15 | 0.007 | 15 |
| | | Reduction of Hospitalization Days | 0.004 | 16 | 0.011 | 16 | 0.005 | 16 |

## 5. Discussion

It describes the evaluation tools of social enterprises developed to assess the feasibility of business models of sustainable social enterprises and to measure the socio-economic value of corporate activities. Through this manual, social entrepreneurs can establish business plans with high socio-economic value and business attractiveness, and social enterprise support organizations can use them as guidelines for identifying and supporting social enterprises with high potential for success.

### 5.1. Social Enterprises Evaluation Model Using SROI

The following indicators model can be used for the evaluation of sustainable social enterprises presented in this study. Absolute value can be assessed through SROI, but there is a limit to assessing relative value, so it can be used as an indicator for evaluation. By describing both the absolute value and the relative value of a social enterprise, a comparison can be made between them.

### 5.2. Diagnostic Kit on Management

For continuous monitoring of social enterprises, an assessment of the situation of social enterprises should be carried out. However, these tasks are not often used due to the difficulty of collecting and evaluating data. By using the above AHP, a diagnostic kit on management of social enterprises can be created by using only a few important indicators, and a list of companies that need to be managed and the evaluation cycle for these companies are made more frequently than other companies. Although social enterprises have different weights depending on the type, AHP results show that employment, income, and community contributions are the most important at the 2nd level. A diagnostic kit on management by type of social enterprises shown in Table 5 below is prepared using the weighted percentages of the AHP results for three sectors at Level 2 and six sectors at Level 3.

**Table 5.** Diagnostic Kit on Management by Type of Social Enterprises.

| Social Value & Items | | | Formula | Weights | | |
|---|---|---|---|---|---|---|
| Level 1 | Level 2 | Level 3 | | Employment-Type | Social Service-Type | Mixed-Type |
| Employment & Income Increase | Employment | Newly Hired Personnel | (Income Levels in Social Enterprise −Income Levels of Previous Work) | 0.658 | 0.362 | 0.469 |
| | | Social Work Participants' Switch to Similar Work After Contract Expiration | (Current Income Level for Work −Income Level for Work Place Before Social Enterprise Employment) | 0.129 | 0.193 | 0.126 |
| | Income | Income Increase for Vulnerable Workers | (Income Levels in Social Enterprise − Income Levels of Previous Work) | 0.104 | 0.117 | 0.179 |
| | | Income Increase of Workers'/Service Users' Family Through Economic Activities | (The Number of Members Who Became Economically Active by Using the Service) ×(Working Hours) × (Minimum Wage) | 0.023 | 0.067 | 0.079 |
| Community Contribution & Govn't Budget Reduction | Community Contribution | Affordable Social Services | (Market Price of Social Service −Price of Social Service Provided by the Company) × Number of Service Provided | 0.057 | 0.127 | 0.093 |
| | | Free Provision of Social Services | (Applicable Market Price) × (Frequency of Service Provided) | 0.029 | 0.134 | 0.054 |

The social value index for the social value evaluation of employment-type and social service-type enterprises can be summarized as Appendix A (Table A1. Social Value Items). It explains how social value is measured and describes the source, amount, duration and monetary approximation of the data. It, that is, measures the social value of the enterprise and evaluates the degree of change, such as measuring the difference between the market value and the offered value of the social enterprise. As well, the social value of the enterprise is measured using SROI and changes are monitored. Based on this, SROI can be applied to assess socio-economic values by sector (Appendix B: Table A2. Socio-economic Value Assessment by Sector), year (Appendix C: Table A3. Socio-economic Value Assessment by Year) and type (Appendix D: Table A4. Social Value Assessment by Type).

## 6. Conclusions

Major conclusion from this research as follows: First, in order to prepare a plan for evaluating social enterprises by applying SROI, an SROI application evaluation model was proposed. In order to prepare the evaluation indicators, the Delphi survey was used to derive the evaluation index items suitable for the social enterprise to the experts in the first and second survey, and the weights were calculated through AHP analysis. The results of the Delphi and AHP analysis showed that employment was the most important factor in social enterprises in Korea, with the highest share of newly hired personnel. Employment-type social enterprises have the highest priority in terms of employment, income (income increase for vulnerable workers), and community contribution (affordable of social services), while the social service type was in order of employment, community contribution, and income (income increase for the vulnerable workers). On the other hand, the mixed type was employment (newly hired personnel), income (income increase of vulnerable workers), employment (social work participants' switch to similar work after contract expiration), and community contribution (affordable of social services).

Second, this model can be used for evaluation of social enterprises and evaluation of Ministry of Employment and Labor in Korea certified social enterprises. The evaluation of social enterprises by applying SROI, which is the result of this study, will enable more accurate valuation of social enterprises and suggest clear results due to evaluation indicators based on quantitative evaluation. It can also be provided as a basis for individuals, companies and investment funds in investing in social enterprises that are common in developed countries. In addition, it can be used as a data to promote the importance of social enterprises through accurate evaluation of their socioeconomic value.

In summary, although there are various advantages and disadvantages for each evaluation in measuring the performance of social enterprises, it is necessary to choose a valuation method that is commonly used, if the purpose of valuation is to manage the social enterprises. Although social enterprises are generally in their infancy, they show more socio-economic value than government subsidies, even though they are not exactly valued in terms of providing social services or employment. This social value is expected to increase as social enterprises' service stabilization progresses.

Considering these results, in order to increase the value of social enterprise, it is necessary to make efforts to form social capital by raising the public's awareness of social value with efficient management through various evaluations of social enterprises and the emergence of various social enterprises. It is also advantageous to select a method of value measurement in connection with government policies. At present, the government in Korea is trying to measure the value of social enterprises, and it is expected that detailed methods will be suggested through further research. In the future, the evaluation of social enterprises to be studied together with the valuation of social enterprises that applied SROI will be able to efficiently cultivate and manage social enterprises in Korea. As well, this research emphasizes the need to better understand social enterprises as a multi-scholar and multi-dimensional organization that includes a multi-faced mechanism of social, economic, and environmental community development, away from understanding social enterprises as a specific business model.

This research has few limitations despite of the findings. The model presented in this study has limitations in accurately measuring the value of social enterprises in the existing financial analysis of return on investment. Subsequent research requires the continuous development of a new model that can properly judge these values while the positive externalities, which are the main characteristics of social enterprises, are highlighted. Although the possibility of generalization is limited, this research suggests an agenda for future research in this area and suggests that our conceptual framework has a broader scope beyond Korea's context, as it is derived from a wider international literature.

**Author Contributions:** Conceptualization, D.-J.K. and Y.-S.J.; methodology, D.-J.K. and Y.-S.J.; software, D.-J.K.; validation, Y.-S.J.; formal analysis, D.-J.K.; investigation, D.-J.K.; resources, D.-J.K.; data curation, Y.-S.J.; writing—original draft preparation, D.-J.K. and Y.-S.J.; writing—review and editing, Y.-S.J. All authors have read and agreed to the published version of the manuscript.

**Funding:** This research received no external funding.

**Acknowledgments:** This paper was sponsored by Woosuk University, Korea. The efforts from the reviewers into making positive suggestions and helpful comments to improve the paper are greatly appreciated. Its contents are solely the responsibility of the authors and do not necessarily represent the official views of Woosuk University.

**Conflicts of Interest:** The authors declare no conflict of interest.

## Appendix A

**Table A1.** Social Value Items.

| | Items | | Formula | Result ($) |
|---|---|---|---|---|
| | **Social Return** | | **Social Benefits − Social Costs** | - |
| **Social Benefits** | Employment | Newly Hired Personnel | Income Levels in Social Enterprises − Income Levels in Previous Work | - |
| | Income | Income Increase for Vulnerable Workers | Income Levels in Social Enterprises − Income Levels in Previous Work | - |
| | Self-development | Technical Competence Through Vocational Activities | Training Costs of Vocational Skills | - |
| | Self-esteem | Family Counseling & Free Education | Family Counseling (Education) Fees × Number Provided | - |
| | Community Contribution | Affordable Social Services | (Market Price of Social Service − Social Service Prices of Enterprises) × Number of Service Provided | - |
| | | Free Provision of Social Services | Market Price of Social Service × Number of Service Provided | - |
| | Government Budget Reduction | Budget Saving Through Consignment Management of Social Welfare Services | (Direct Operating Costs − Costs of Consignment on Social Enterprises) × Scale of Consignment | |
| | Safety & Increase Health Level | Reduction of Safety Accidents in Social Enterprises | (Average Number of Safety Accidents in General Facilities − Number of Safety Accidents at the Facility) × Compensation Insurance per Person and Accident Insurance Coverage | - |
| | Total | | | - |
| **Social Costs** | Government Subsidies | | Labor Costs + Business Development Costs + Others | - |

## Appendix B

**Table A2.** Socio-economic Value Assessment by Sector.

| | Items | | Sector (e.g.) | | | | Result ($) |
|---|---|---|---|---|---|---|---|
| | | | Care | Manufacturing | Environments | Others | - |
| | **Social Return** | | - | - | - | - | - |
| **Social Benefits** | Employment | Newly Hired Personnel | - | - | - | - | - |
| | | Social Work Participants' Switch to Similar Work After Contract Expiration | - | - | - | - | - |
| | Income | Income Increase for Vulnerable Workers | - | - | - | - | - |
| | | Income Increase of Workers'/Service Users' Family Through Economic Activities | - | - | - | - | - |
| | Self-development | Certification Through Vocational Activities | - | - | - | - | - |
| | | Technical Competence Through Vocational Activities | - | - | - | - | - |
| | | Free Providing Social Training for Workers | - | - | - | - | - |
| | Self-esteem | Family Counseling & Free Education | - | - | - | - | - |
| | | Free Providing Cultural Programs for Workers | - | - | - | - | - |
| | Community Contribution | Affordable Social Services | - | - | - | - | - |
| | | Free Provision of Social Services | - | - | - | - | - |
| | Government Budget Reduction | Budget Reduction Through Consignment Management of Social Welfare Services | - | - | - | - | - |
| | Safety & Increase Health Level | Reduction of Safety Accidents in Social Enterprises | - | - | - | - | - |
| | Total | | - | - | - | - | - |
| **Social Costs** | Government Subsidies | | - | - | - | - | - |

## Appendix C

**Table A3.** Socio-economic Value Assessment by Year.

| Items | | | Year (e.g.) | | | | Result ($) |
|---|---|---|---|---|---|---|---|
| | | | 2015 | 2016 | 2017 | 2018 | - |
| **Social Return** | | | - | - | - | - | - |
| **Social Benefits** | Employment | Newly Hired Personnel | - | - | - | - | - |
| | | Social Work Participants' Switch to Similar Work After Contract Expiration | - | - | - | - | - |
| | Income | Income Increase for Vulnerable Workers | - | - | - | - | - |
| | | Income Increase of Workers'/Service Users' Family Through Economic Activities | - | - | - | - | - |
| | Self-development | Certification Through Vocational Activities | - | - | - | - | - |
| | | Technical Competence Through Vocational Activities | - | - | - | - | - |
| | | Free Providing Social Training for Workers | - | - | - | - | - |
| | Self-esteem | Family Counseling & Free Education | - | - | - | - | - |
| | | Free Providing Cultural Programs for Workers | - | - | - | - | - |
| | Community Contribution | Affordable Social Services | - | - | - | - | - |
| | | Free Provision of Social Services | - | - | - | - | - |
| | Government Budget Reduction | Budget Reduction Through Consignment Management of Social Welfare Services | - | - | - | - | - |
| | Safety & Increase Health Level | Reduction of Safety Accidents in Social Enterprises | - | - | - | - | - |
| | | Total | - | - | - | - | - |
| **Social Costs** | | Government Subsidies | - | - | - | - | - |

## Appendix D

**Table A4.** Social Value Assessment by Type.

| Items | | | Type | | | Result ($) |
|---|---|---|---|---|---|---|
| | | | Employment | Social Service | Mixed | - |
| **Social Return** | | | - | - | - | - |
| **Social Benefits** | Employment | Newly Hired Personnel | - | - | - | - |
| | | Social Work Participants' Switch to Similar Work After Contract Expiration | - | - | - | - |
| | Income | Income Increase for Vulnerable Workers | - | - | - | - |
| | | Income Increase of Workers'/Service Users' Family Through Economic Activities | - | - | - | - |
| | Self-development | Certification Through Vocational Activities | - | - | - | - |
| | | Technical Competence Through Vocational Activities | - | - | - | - |
| | | Free Providing Social Training for Workers | - | - | - | - |
| | Self-esteem | Family Counseling & Free Education | - | - | - | - |
| | | Free Providing Cultural Programs for Workers | - | - | - | - |
| | Community Contribution | Affordable Social Services | - | - | - | - |
| | | Free Provision of Social Services | - | - | - | - |
| | Government Budget Reduction | Budget Reduction Through Consignment Management of Social Welfare Services | - | - | - | - |
| | Safety & Increase Health Level | Reduction of Safety Accidents in Social Enterprises | - | - | - | - |
| | | Total | - | - | - | - |
| **Social Costs** | | Government Subsidies | - | - | - | - |

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
