# Peer review of "The Evaluation Model on an Application of SROI for Sustainable Social Enterprises"

_2199-8531, doi:10.3390/joitmc6010007_

Round 1
Reviewer 1 Report
The topic of the study is in line with the scope of the journal. The subject is also considered significant considering the increasing importance of sustainability in social enterprises. However, the paper needs to have some modifications related to its explanation and structure. For example, the introduction section needs to be rewritten and Literature Review needs to be provided. The summary of the comments are listed as follows.
In the abstract, please provide the problem statement that the study wants to solve. The abbreviation (SROI) should be initially defined. For the text clarity would you refrain from using additional words, mostly meaningless filler words, which can be omitted or some archaic words see e.g. “respectively”, “thus”, “hence”, therefore”, “furthermore”, “thereby”, “basically,”, “meanwhile”,” wherein”, “herein”, “hitherto”, “Nonetheless”, “Perceivably”, “whereas”, etc.? The explanation contents many unnecessary statements. It needs to be rewritten. In the new introduction, you need to focus sharply on what motivates your choice of the topic for your study, especially the importance of the issue and the major gaps in the literature. You need to explicitly state the specific purpose of your study, especially where you seek to make contributions. Also, you need to show how you plan to achieve your purpose as well as the specific contributions you have achieved and research novelty. Please eliminate the multiple references. After that please check the manuscript thoroughly and eliminate all the lumps in the manuscript. This should be done by characterising each reference individually. This can be done by mentioning 1 or 2 phrases per reference to show how it is different from the others and why it deserves mentioning. Please provide a brief explanation between section and sub-section, e.g. Section 2 and 2.1. Literature Review section is required to deeply review the previous related studies and reveal the knowledge and inconsistencies in the literature, then relate them with the research objective. Table 5 is unorganised. Some texts are missing. Please modify it. Please re-arrange the reference list. Remove no. 1 in line 2. For books, please add the country.Author Response
Dear Sir/Madam,
I tried to do my best revise this paper.
Please see the attachment.
Best regards,
YS

Reviewer 2 Report
Seems that the title is slightly misleading. The paper prepares an application of SROI, not focusing on that.
The article is partly troublesome to read. The authors do not explain / disclose all terms they use, e.g., total score (p. 4) - which score, how calculated; social value index (p. 5); λ and n in formula (1). It is good practice to give explanations of terms and symbols in the footer of the table and after presenting the formula (I recommend to follow). Partly this also is/seems due to very laconic writing. It would be helpful to have (presentation of) Social Value Items matrix in the appendixes of the article.
The formulas in Table 5 are incomplete (in the version I got) and therefore poorly legible. Also, the Diagnostic Kit could be further explained.
In addition, AHP author Saaty's name is misspelled in the references list. Also, I couldn't find reference to him in the text.
Author Response
Dear Sir/Madam,
I tried to do my best to revise this paper.
Please see the attachment.
Best regards,
YS

Round 2
Reviewer 1 Report
Thank you for the modifications made in accordance with the reviewers' comments. The current paper version can be published as is.